# Modified Early Warning Score: Clinical Deterioration of Mexican Patients Hospitalized with COVID-19 and Chronic Disease

**DOI:** 10.3390/healthcare11192654

**Published:** 2023-09-29

**Authors:** Nicolás Santiago González, María de Lourdes García-Hernández, Patricia Cruz-Bello, Lorena Chaparro-Díaz, María de Lourdes Rico-González, Yolanda Hernández-Ortega

**Affiliations:** 1Hospital Regional de Alta Especialidad Ixtapaluca (HRAEI), Universidad Autónoma del Estado de México (UAEMex), Ixtapaluca 56530, Mexico; nicosantiago21@hotmail.com; 2Facultad de Enfermería y Obstetricia, Universidad Autónoma del Estado de México (UAEMéx), Toluca 50000, Mexico; patriciacruzbello@yahoo.com.mx (P.C.-B.); lulu78rich@gmail.com (M.d.L.R.-G.); yola.ho@gmail.com (Y.H.-O.); 3Nursing Department, Faculty of Nursing, Universidad Nacional de Colombia, Sede Bogotá, Bogotá 111321, Colombia; olchaparrod@unal.edu.co

**Keywords:** COVID-19, clinical deterioration, early warning score, mortality, intensive care unit, vital signs

## Abstract

The objective was to evaluate the Modified Early Warning Score in patients hospitalized for COVID-19 plus chronic disease. Methods: Retrospective observational study, 430 hospitalized patients with COVID-19 and chronic disease. Instrument, Modified Early Warning Score (MEWS). Data analysis, with Cox and logistic regression, to predict survival and risk. Results: Of 430 patients, 58.6% survived, and 41.4% did not. The risk was: low 53.5%, medium 23.7%, and high 22.8%. The MEWS score was similar between survivors 3.02, *p* 0.373 (95% CI: −0.225–0.597) and non-survivors 3.20 (95% CI: −0.224–0.597). There is a linear relationship between MEWS and mortality risk R 0.920, ANOVA 0.000, constant 4.713, and coefficient 4.406. The Cox Regression *p* 0.011, with a risk of deterioration of 0.325, with a positive coefficient, the higher the risk, the higher the mortality, while the invasive mechanical ventilation coefficient was negative −0.757. By providing oxygen and ventilation, mortality is lower. Conclusions: The predictive value of the modified early warning score in patients hospitalized for COVID-19 and chronic disease is not predictive with the MEWS scale. Additional assessment is required to prevent complications, especially when patients are assessed as low-risk.

## 1. Introduction

In Mexico, the first case of COVID-19 was registered on 28 February 2020, and it was not until 9 May 2023 before the end of the COVID-19 health emergency was announced, where 7,595,863 cases and 333,961 deaths were confirmed [1]. In 2021, there was an excess of mortality through estimates of endemic channels of 47.2%, compared to the previous five years, placing COVID-19 as the main cause [2]. COVID-19 added to chronic diseases such as diabetes, high blood pressure, obesity, and chronic kidney damage, increased mortality in patients with COVID-19. Likewise, age over 60 contributed to the risk of death [3]. Similarly, the triad of hypertension, hyperlipidemia, and obesity had a significant effect as risk factors for complications, although their impact was slight on mortality [4]. One study demonstrated that cardiac dysfunction is a risk factor because half of the patients observed with severe COVID-19 exhibited diastolic dysfunction of the left ventricle, significantly associated with mortality related to prolonged oxygen support, as it increases the probability of intravascular volume depletion due to fever-induced sweating and a deficit in water intake [5].

Furthermore, a substantial percentage of patients ventilated for COVID-19-induced acute respiratory distress syndrome presented right ventricular involvement, specifically, the acute cor pulmonale phenotype, which increased mortality in various intensive care units [6]. Patients with COVID-19 and chronic disease were more likely to develop severe symptoms upon admission to the intensive care unit and faced a higher mortality risk [7]. A cohort study found that patient demographics and comorbidities were associated with greater clinical severity [8]. 

Therefore, given the high mortality rate during the COVID-19 pandemic, the Modified Early Warning Score (MEWS) for clinical deterioration was used to analyze the predictive value of mortality through the modified early warning score for clinical deterioration in adult patients hospitalized for severe COVID-19. It is essential to mention that the scale is a guide for rapid evaluation in the prediction of hospital mortality and the identification of the high-risk group in patients with COVID-19 [9]. It can be used in triage for early evaluation of severity, considering that the mortality rate from COVID-19 was higher in the geriatric age group and in patients with multiple comorbidities [10]. The MEWS scale is considered effective for predicting 28-day mortality in patients requiring hospitalization for COVID-19 [11]. Although using the Rapid Emergency Medicine Score (REMS) can provide adjunctive risk stratification in critically ill patients with COVID-19 [12], death in COVID-19 patients with chronic disease, in addition to symptoms or signs neurological, was predicted by older age, malignancy, and higher MEWS scores on admission [13]. Older age, low oxygen saturation, and decreased lymphocytes had a high risk of COVID-19-related mortality. However, one study revealed that REMS has better prognostic ability than MEWS [14]. The advantage of MEWS is that it is obtained by measuring vital signs and allows risk stratification in the clinical context of each patient [15]. A MEWS score of 5 or higher is associated with a higher probability of admission to an intensive care unit or death [16].

There are various traditional clinical prediction scales that help predict the risk of mortality from COVID-19. However, during the COVID-19 era, computational prediction methods have been used with the same objective. Not only are MEWS indicators evaluated, but also environmental factors. For example, the COVID-19 ML machine-learning model analyzes the correlation between contaminants circulating in the air and the number of infected people [17]. Likewise, the methodology called COVID-19 Community Temporal Visualizer (CCTV) analyzes the impact of the clinical evolution of the pandemic based on clinical, climatic and contamination data in different areas and times; for example, the number of patients with COVID-19 in various Intensive Care Units, facilitating the prediction of mortality [18]. Therefore, advanced technologies must be used to predict and prevent the clinical deterioration of patients hospitalized with COVID-19 and chronic diseases. 

## 2. Materials and Methods

### 2.1. Setting

It is a retrospective observational study in a High Specialty Hospital in Mexico in 430 adult patients with severe COVID-19, hospitalized between 1 July and 30 December 2021, in the Intermediate and Intensive Care Unit, who were assessed for the risk of clinical deterioration. The MEWS scale was not followed up. The information was extracted from electronic nursing records of the hospital’s clinical file. The study criteria were records of men and women over 16 years of age with chronic disease and a positive result for SARS-CoV-2 using the in-situ polymerase chain reaction test, type of discharge, and hospital mortality. Severe COVID-19 infection was defined as an oxygen saturation level of less than 90% and needing supplemental oxygen administration. Pediatric patients, pregnant women, people diagnosed with chronic diseases of respiratory origin, and those who had restricted access to the file or the data were incomplete were excluded.

### 2.2. Data Collection and Measurements

The measurement instrument used was the Modified Early Warning Score (MEWS) for Clinical Deterioration [16]. The MEWS score was assessed once, upon the patient’s admission to the intensive care unit, and subsequently at 30 days, where the survival rate of patients and the type of hospital discharge were reviewed. The data collected included Systolic Blood Pressure (SBP), Heart Rate (HR), Respiratory Rate (RR), Body Temperature (T), Level of Consciousness (AVPU), Oxygen Therapy, and Oxygen Saturation (SpO2). With this data, the risk of mortality (low, medium, and high) survival is calculated, and it can be used in all hospitalized patients for early detection of clinical deterioration and the possible need for a higher level of care. The scores obtained are evaluated on a scale from 0 to 14 points. A score ≥5 is statistically related to a greater probability of death or admission to an Intensive Care Unit (ICU); from 1 to 2 points (low risk) is equivalent to a 7.9% probability of admission to the ICU or death within 60 days; 3 to 4 points (medium risk) means 12.7% and having 5 to 14 points (high risk) increases the mortality rate to 30%. The technique used to calculate the MEWS index for each patient was a spreadsheet to identify the rate of patients at high risk and hospital mortality. Considering the type of discharge of the patients, they were classified as survivors and non-survivors.

### 2.3. Statistic Analysis

Data are presented in percentages for categorical variables and medians for continuous variables. The significant association between survival and the MEWS value was studied using Pearson’s Chi-square test, and the Kolmogorov-Smirnov test was performed to assess the normality of the numerical variables. The one-sample *t*-test was used to assess the mean MEWS score, and the independent-sample *t*-test was used to compare survival. Risk estimation of the MEWS index was made using odds ratio (OR) and relative risk (RR). Kaplan-Meier survival analysis, followed by Cox regression analysis, was run to predict the survival and hazard functions. A simple and multiple linear regression model of the MEWS score was developed for mortality risk and clinical deterioration in survivors and non-survivors. Statistical significance was considered with the value *p* < 0.05. Version 22 of the IBM SPSS statistical package was used. The study complies with the guidelines of the Declaration of Helsinki and was approved by the Research and Ethics Committee at the Hospital Regional de Alta Especialidad de Ixtapaluca.

## 3. Results 

Four hundred and thirty (430) hospitalized patients with COVID-19 and chronic disease were included: diabetes stands out at 51.5%, arterial hypertension at 29.8% and cardiovascular disease at 14%. 44% (189) were women and 56% (241) were men, with a mean age of 55.2 years (SD 15.3). 58.6% (252) survivors and 41.4% (178) non-survivors.

The modified early warning score is presented in Figure 1. A range of zero to twelve points was identified. The MEWS score was stratified into three risk groups: (1) low-risk group with 53.5% (230) of patients presenting from one to two points, equivalent to a 7.9% probability of admission to the (ICU) or death within 60 days. Therefore, it should be continued providing health care; (2) the medium-risk group, 23.7% (102) obtained three to four points related to a 12.7% probability of admission to the ICU or death; and the (3) high-risk group, with 22.8% (98) was the most affected group, with 5 to 12 points with 30% high risk of clinical deterioration, for which a higher level of care and health care was considered.

Table 1 describes the risk of mortality and the value MEWS between the group survival and did not survival, shows that 41.4% (178) of the study population did not survive 30 days, in whom their MEWS score was previously evaluated as low risk 20.9%, medium risk 10.9%, and high risk 9.5%. The highest percentage of patients who died were evaluated with low MEWS. No statistical difference in the total percentage was found between the medium and high MEWS. In Fisher’s exact test, men had higher MEWS than women. In the relationship between survival and the MEWS value, the chi-square test with 1.046, the *p*-value of 0.307, is above the significance level. That is, there is no relationship or significant difference between both groups.

The mean oxygen saturation of the patients was 89.5% (SD 13.5), 36% (155) presented hypoxia (oxygen saturation < 90%) and 64% (275) presented normal saturation, with supplementary oxygen support, which is a protective factor for patients with respiratory distress syndrome, for which the oxygenation devices used are described. 20.9% had invasive mechanical ventilation and 0.9% tracheostomy, 28.8% had a low-flow nasal catheter, 15.6% had a high-flow nasal catheter, 17.9% had a reservoir mask, and 4.4% had a simple mask. Of the patients with hypoxia, 42.6% (66) presented low MEWS, and 57.4% (89) had high MEWS.

In the retrospective analysis of the odds ratio (OR), the high MEWS (exposure factor) is a precedent for non-survival; in the risk estimation, it was significant with an OR value of 0.818 (95% confidence interval [CI], 0.557–1.202).

Regarding the relative risk (RR) in the cohort design, regarding the probability of presenting hypoxia in the follow-up study, it was found that 28.7% of people with low MEWS presented hypoxia, while 44.5% of people with MEWS high still developed hypoxia. It is significant in the Chi-square test with a value of 11.59 and *p* = 0.001. In the risk estimation measures, I present a value of 1.285 (95% CI, 1.107–1.491). Therefore, it is considered significant that the probability of presenting hypoxia is greater in patients with high MEWS than in patients with low MEWS.

Table 2 shows the comparison of means. In the *t*-test for a sample, when evaluating the mean of the MEWS score of the study population, 3.09 (SD 2.135) was obtained, with a *p*-value of 0.000 (95% CI, 2.89–3.30), so the value of the study mean is different from the population mean of 0.0, so there is a significant difference. In the *t*-test for independent samples, the survivors presented MEWS of 3.02 (DE 2.142) and non-survivors of 3.20 (DE: 2.127). When comparing the means, *t* = 0.891 and *p* 0.373 (95% CI −0.225, 0.597), the null hypothesis is accepted since there is no statistical significance, that is, there is no difference in the means of the survivors concerning the MEWS variable of the groups compared.

Table 3 describes the simple and multiple linear regression model. In the simple linear regression model on the level of oxygen saturation and the MEWS score, an R of 0.432 and ANOVA with a *p*-value of <0.000, with a constant of 9.181 and coefficient for the predictor variable oxygen saturation of −0.068, therefore, if there is a linear relationship between both variables.

36% of patients presented hypoxia despite having supplemental oxygen. That is, their oxygen saturation levels decreased <90%. Therefore, 21.8% invasive and 78.2% non-invasive oxygenation devices were used to help reverse hypoxia. When performing statistical analysis, the multiple linear regression model exhibited an R of 0.920 and to confirm the data in the linear regression model, the ANOVA was applied with a significant *p*-value of 0.000, MEWS score constant 4.713, the coefficients were: oxygen saturation −0.037, diabetes 0.059, hypertension 0.161, patient sex 0.078, use of oxygenation device 0.026, medium risk of mortality 1.845, high risk of mortality 4.406 and presence of hypoxia −0.271. No statistical correlation was found between COVID-19 infection and diabetes mellitus (*p* 0.472), nor with systemic arterial hypertension (*p* 0.073).

A Cox regression analysis was developed to predict the survival and hazard function. In the Kaplan-Meier survival analysis, a median survival value in days was identified, 32.11 days of hospitalization (95% CI 22.97–36.25). Subsequently, three predictors were used: the risk of clinical deterioration, the use of oxygen, and invasive mechanical ventilation. For the creation of the Cox Regression model, obtaining the tests for the model coefficients, the *p*-value is 0.011, significant, meaning that it does fit the model. Observing the coefficients for each variable, the coefficient for the risk of clinical deterioration is 0.325, a positive one. The coefficient for invasive mechanical ventilation has a negative value of −0.757. Subsequently, for the risk coefficients, the risk of clinical deterioration is 1.384, above one and positive, while the coefficient for invasive mechanical ventilation is 0.469, under unit.

Regarding the prediction, the survival function is presented in Figure 2; at the beginning, all the participants were alive, monitoring the time in days, and at the end of the study, 41.4% (178) had died. It is worth mentioning that the aim is to predict the survival function. 

The cumulative risk function is expressed in Figure 3, where at the beginning of the study, the risk is one, and this risk increases as time increases in days until it almost has a value of 2.5. With this model, we sought to predict the risk function of the patients.

## 4. Discussion

The Modified Early Warning Score (MEWS) predicts mortality risk and ICU admission. The study observed that the probability of clinical deterioration was high in approximately half of adult patients hospitalized for COVID-19 plus chronic disease, in whom a higher level of health care should be considered, according to the MEWS score. The mortality rate was high, with 41.4% (178) non-survivors. There was no difference in the MEWS score between the low versus medium and high-risk groups for clinical deterioration since non-survival similarly affected them. MEWS does not appear useful for predicting 30-day mortality in geriatric patients with COVID-19 disease [19].

The analysis between survival and the MEWS value did not show a relationship or significant difference between survivors and non-survivors. The high MEWS score was similar in both groups, unlike the results of Sub-Na et al. (2022), who identified a higher MEWS value in the non-survivor group than in the survivor group [20]. Therefore, MEWS could only be useful to identify patients with a low risk of clinical deterioration [21]. Leszek et al. (2022) identified that patients who died during hospitalization were associated with higher age, prevalence of diabetes, and high MEWS score [13].

More than half of the patients who died were evaluated with a low MEWS. It would be expected that they would have had a high MEWS, considering that the higher the score, the greater the risk. Therefore, it is suggested to use another additional assessment, such as the Sequential Assessment of Organ Failure (SOFA) score, because it showed better performance in prediction versus MEWS. Therefore, it is considered an efficient tool to assess hospital mortality in older adult patients with severe COVID-19 [20]. Similarly, Hu et al. point out that the efficacy of REMS for screening these patients is attributed to its high negative predictive value [14]. The SOFA clinical score was more accurate than MEWS in predicting the need for intensive care and mortality [22]. EWS scores warn of clinical deterioration. However, evidence shows that the scores fail to predict 30-day mortality [23].

Half of the patients had low MEWS, similar to the results of Covino et al. [21], who described that MEWS had the lowest overall precision, so the early warning score could be useful to identify patients at low risk of clinical deterioration, but not, with those of medium and high risk. Jawad et al. (2023) reported that MEWS was a poor predictor of 28-day mortality. The low sensitivity in predicting ICU admission could be improved by combining MEWS with the clinical assessment [24]. One study showed that the NEWS was more accurate than the MEWS in identifying mortality risk [25]. 

Diabetes and hypertension were the main comorbidities. It should be noted that arterial hypertension and infectious processes increase the MEWS score when presenting fever, tachycardia, and hypertension, so it is recommended to use this score to predict the need for specific care [26]. In another study, hypertension was significantly associated with mortality in patients with COVID-19 [10]. Men were more affected, with whom diabetes predominated as the main comorbidity, with results similar to those of Singh [27], who points out a significant increase in the severity and mortality of COVID-19 in people with diabetes mellitus. Likewise, arterial hypertension predominated. Although it is a single-center study, morbidity rates are by national statistics [1]. The age variable affected people aged 55 on average, unlike the findings of Wang et al. (2020), who obtained a positive coefficient that Implies that the group of male patients and those older than 75 had a higher risk of death. MEWS values efficiently predict hospital mortality in elderly patients with COVID-19 [9].

Logistic regression models are the standard for developing prediction models. In our results, the Cox regression model showed statistical significance for each coefficient (risk of clinical deterioration, use of oxygen, and use of invasive mechanical ventilation). The risk of clinical worsening had a positive coefficient, meaning that a higher risk of clinical deterioration is associated with higher mortality. The coefficient for invasive mechanical ventilation (IMV) is negative because providing oxygen through IMV is a protective factor. When patients receive oxygen and IMV, it favors survival and, consequently, low mortality. Regarding the survival and cumulative risk function, significant results were obtained, similar to that reported by Hai Hu et al. (2021), where they demonstrated a statistically significant ability to predict mortality in patients with COVID-19 through regression MEWS logistics [14].

MEWS is oriented towards prediction; however, more clinical implications that intervene in the evolution of the disease and that determine its outcome must be considered. The MEWS and TREWS scores were effective in predicting 28-day mortality in patients requiring hospitalization for COVID-19 [11]. The MEWS scale identified a higher prediction over the NEWS score; therefore, it could be considered an efficient prediction score for morbidity and mortality in critically ill patients [28]. One study demonstrated that MEWS is more accurate than qSOFA in predicting patient admission to the ICU [29].

In other clinical settings, MEWS showed good performance in predicting in-hospital mortality in adult patients with non-traumatic emergencies [30] and in traumatic emergencies [31]. MEWS had better predictive efficacy than the revised trauma score (RTS) [32]. The Circulation, Respiration, Abdomen, Motor, and Speech Score (CRAMS) was superior to the MEWS for predicting severity in polytrauma patients [33]. The MEWS-Abdomen is superior to the MEWS in the assessment of severe trauma [34]. The different contexts showed the usefulness of the MEWS evaluation. However, in most cases, an additional evaluation is included in the standardized parameters in the MEWS scale. For this reason, it is recommended to include capillary glucose levels and oxygen saturation during the evaluation of the patient affected by COVID-19 plus chronic disease. 

The predictive value of continuous measurement of vital signs is of great value for adequate control [35]. MEWS is more accessible to perform and generally applicable [36]. A MEWS score > 5 at 24 h after admission was significantly associated with in-hospital mortality, so the application of the MEWS can be a useful decision-making tool [37]. Using eMEWS helped to identify hospitalization and mortality [38] more accurately. 

Limitations. A retrospective design was used in a single center and with a relatively low number of patients. Only the MEWS scale was used, which limited its comparison with other predictors. The hospitalization of patients in the ICU depended on various factors and not on the prediction of our results. Therapeutic interventions and clinical outcomes were not considered. Therefore, the factors that modify vital signs in the MEWS score are unknown.

Suggestions. Prospective multicenter studies using various prediction scales are recommended to analyze external validity and enable the generalization of results.

## 5. Conclusions

The MEWS value was low on the likelihood of clinical deterioration in hospitalized adult patients with chronic disease and COVID-19, yet mortality rates were high. More than half of the patients with a predictive result using the MEWS scale in patients who died had a low score. Therefore, constant monitoring of vital signs is required in primary care. The strategy for caring for patients with COVID-19 disease must prioritize patients with comorbidities, not only making predictions but also carrying out nursing interventions that reduce risks. Finally, the MEWS scale is not valid for detecting clinical deterioration and identifying a higher level of care in this group of patients.

## Figures and Tables

**Figure 1 healthcare-11-02654-f001:**
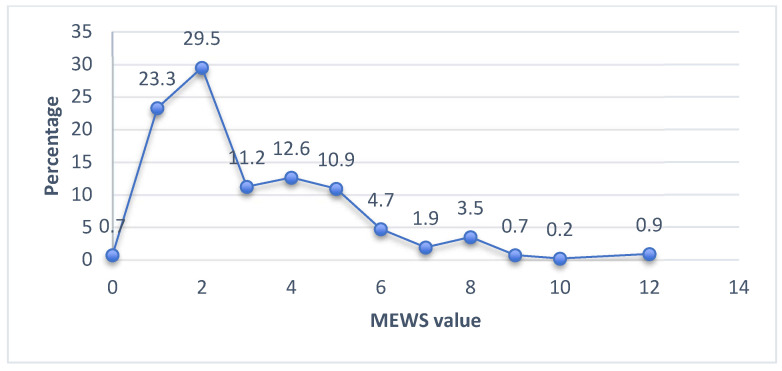
Modified Early Warning Score (MEWS). Source: Instrument “Clinical deterioration in patients with chronic disease and COVID-19, assessed by Modified Early Warning Score”.

**Figure 2 healthcare-11-02654-f002:**
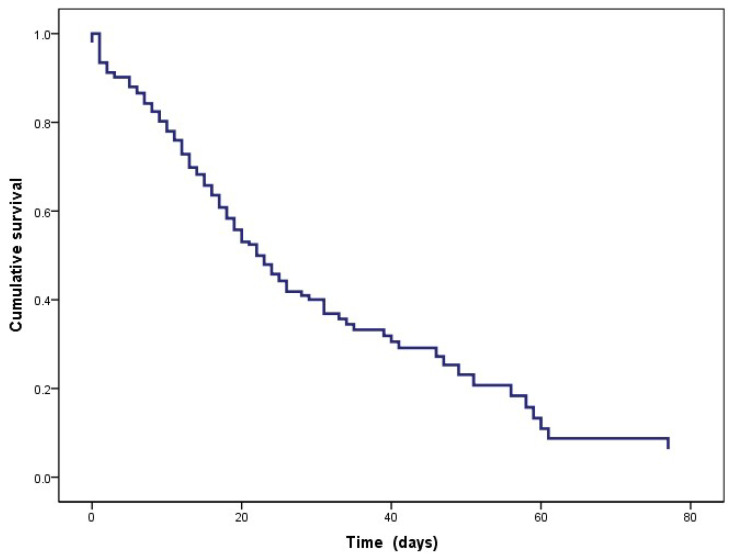
Prediction of the survival function.

**Figure 3 healthcare-11-02654-f003:**
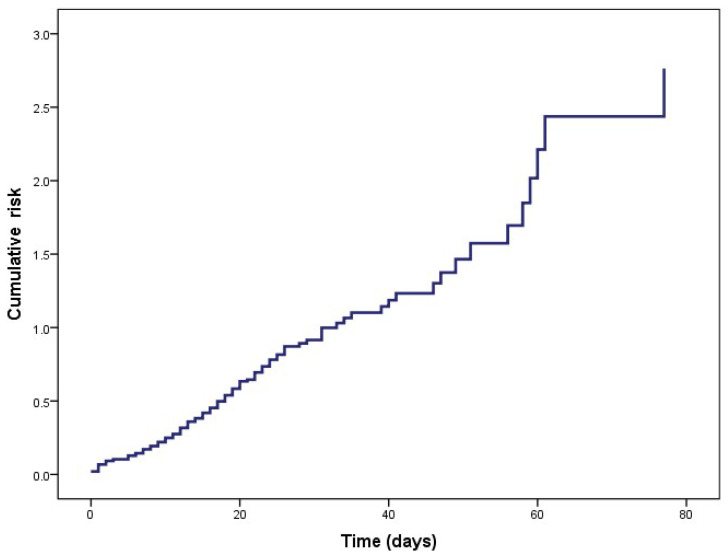
Prediction of the hazard function.

**Table 1 healthcare-11-02654-t001:** Percentage of patients who survived 30 days.

MEWS Value	MRI	Total		Did Not Survive		Survive		Women		Men	
	%	%	No.	%	No.	%	No.	%	No.	%	No.
Low	7.9	53.5	230	20.9	90	32.6	140	24.5	105	29	125
Half	12.7	23.7	102	10.9	47	12.8	55	10.9	47	12.8	55
High	30	22.8	98	9.5	41	13.3	57	8.6	37	14.2	61
Total		100	430	41.4	178	58.6	252	44	189	56	241

Source: Instrument “Clinical deterioration in patients with chronic disease and COVID-19, assessed by Modified Early Warning Score”. MRI: Mortality risk.

**Table 2 healthcare-11-02654-t002:** Comparison of means of the MEWS score.

Test	Score	MEWS Score			Confidence Interval 95%	
		Mean	SD	*p*	Lower	Superior
*t* for a sample	30.03	3.09	2.135	0.000	2.89	3.30
*t* for independent samples: Survivors	0.891	3.02	2.142	0.373	−0.225	0.597
*t* for independent samples: Non-survivors	0.892	3.20	2.127		−0.224	0.597

Source: Instrument “Clinical deterioration in patients with chronic disease and COVID-19, assessed by Modified Early Warning Score”. SD: Standard deviation.

**Table 3 healthcare-11-02654-t003:** Linear regression model of the MEWS score.

Linear Regression Model	Score		MEWS Score			
	R.	ANOVA	Constant	Coefficient	*t*	*p*
Simple linear regression model: Oxygen saturation	0.432	0.000	9.181	−0.068	−9.902	0.000
Multiple Linear Regression Model: Oxygen Saturation	0.920	0.000	4.713	−0.037	−9.783	0.000
Diabetes				0.059	0.719	0.472
Hypertension				0.161	1.799	0.073
Sex				0.078	0.948	0.344
Use oxygenation device				0.026	1.201	0.230
Hypoxia				−0.271	−2.593	0.010
Medium risk of mortality				1.845	17.256	0.000
High risk of mortality				4.406	37.407	0.000

Source: Instrument “Clinical deterioration in patients with chronic disease and COVID-19, assessed by Modified Early Warning Score”. ANOVA: Analysis of variance.

## Data Availability

The data is available in the institution’s clinical file and handled for statistical and research purposes by ethical principles, confidentiality, and the General Health Law on research.

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
