# Peer review of "Modified Early Warning Score: Clinical Deterioration of Mexican Patients Hospitalized with COVID-19 and Chronic Disease"

_healthcare, 2023, doi:10.3390/healthcare11192654_

Round 1

Reviewer 1 Report

I read with interest the manuscript by Gonzales et al on the use on MEWS to predict clinical deterioration of adult patients admitted to a mexican hospital. The study is sound. However, there are some issues that need to be addressed:

- Line 40-42- Authors should also add that a large percentage of COVID-19 patients have been shown to develop cardiac dysfunction, and this findings are associated with an increase in mortality (doi: 10.1111/echo.15462 - doi: 10.1007/s00134-023-07147-z). Please discuss and add these 2 references.

- Methods. Was the study approved by the local ethical committee? Please specify

- Table 2. Please replace "did survived" with "did survive" or "survive".

- Line 133. How did authors define "hypoxia" in this case? Please specify.

- Table 3. Significance is misspelled.

- Results. Line 182-185. Authors should not comment the results presented in this section. Please present the results without explaining them. All the comments should be given in the discussion section.

- Results. The authors should not repeat the data that are presented in the tables, in order not to be redundant and to increase readability. 

- Discussion. Please report the retrospective design, the single center, and the relatively low number of patients as limitations of the study.

Author Response

Thank you very much for taking the time to review this manuscript. Please find the detailed answers below and the corresponding revisions/fixes highlighted/in the track changes in the forwarded files.

Reviewer 2 Report

The authors present a careful and thorough analysis of prognosis of COVID-19 patients using MEWS score. They showed the higher the MEWS score, the worse the prognosis, and the higher the need for ventilator use.

Please include enough information on the following factors. How many times MEWS scores were measured in this study? Was the MEWS score measured only once at admission?

In cases where ICU admission was required, how many days after hospitalization or after onset did the MEWS score increase?

Were there any significant correlations between the degree of elevation and severity and mortality?

Please describe correlations between the prognosis of COVID-19 infection and the severity of combined diabetes and hypertension complicating and its relationship with prognosis be analyzed?

It would be useful for us to see the various complications such as ischemic heart disease, stroke, and emphysema.

On Table 3, there are items called Oxygen Saturation, Oxygen device, and Hypoxia among the items. A device that administers oxygen is used to reduce SaO2. Also, the state of SaO2 depression is hypoxia, and these are considered to be closely related confounding factors. Please describe this interrelationship.

Author Response

(The authors gave the same response as above.)
